# Corals that survive repeated thermal stress show signs of selection and acclimatization

**Orion S. McCarthy**[1]*, **Morgan Winston Pomeroy**[2,3], **Jennifer E. Smith**[1]

**1** Scripps Institution of Oceanography, Center for Marine Biodiversity and Conservation, University of California San Diego, La Jolla, California, United States of America, **2** School of Geographical Sciences and Urban Planning, Arizona State University, Tempe, Arizona, United States of America, **3** Center for Global Discovery and Conservation Science, Arizona State University, Hilo, Hawai'i, United States of America

* omccarth@ucsd.edu

**Data Availability Statement:** All coral growth and bleaching data, as well as R code for analysis, are available from the Dryad database (https://datadryad.org/stash/dataset/doi:10.5061/dryad.9cnp5hqsf).

## Abstract

Climate change is transforming coral reefs by increasing the frequency and intensity of marine heatwaves, often leading to coral bleaching and mortality. Coral communities have demonstrated modest increases in thermal tolerance following repeated exposure to moderate heat stress, but it is unclear whether these shifts represent acclimatization of individual colonies or mortality of thermally susceptible individuals. For corals that survive repeated bleaching events, it is important to understand how past bleaching responses impact future growth potential. Here, we track the bleaching responses of 1,832 corals in leeward Maui through multiple marine heatwaves and document patterns of coral growth and survivorship over a seven-year period. While we find limited evidence of acclimatization at population scales, we document reduced bleaching over time in specific individuals that is indicative of acclimatization, primarily in the stress-tolerant taxa *Porites lobata*. For corals that survived both bleaching events, we find no relationship between bleaching response and coral growth in three of four taxa studied. This decoupling suggests that coral survivorship is a better indicator of future growth than is a coral's bleaching history. Based on these results, we recommend restoration practitioners in Hawai'i focus on colonies of *Porites* and *Montipora* with a proven track-record of growth and survivorship, rather than devote resources toward identifying and cultivating bleaching-resistant phenotypes in the lab. Survivorship followed a latitudinal thermal stress gradient, but because this gradient was small, it is likely that local environmental factors also drove differences in coral performance between sites. Efforts to reduce human impacts at low performing sites would likely improve coral survivorship in the future.

## 1. Introduction

Coral reefs are among the most vulnerable ecosystems on Earth to the impacts of anthropogenic climate change [1–3]. Climate change has increased the severity and frequency of marine heatwaves [4–6], which disrupt the growth and survival of scleractinian corals, the key foundation organisms on shallow water coral reefs throughout the tropics [7]. Marine heatwaves have

**Funding:** Funding was provided to OM by the National Science Foundation (Graduate Research Fellowship Program award; https://www.nsfgrfp.org/), and to JES by the Scripps Family Foundation, the Bohn Family, and other generous donors. Funding was generalized and was not awarded for this project specifically. As such, funders had no role in study design, data collection, analysis, manuscript preparation, or publication.

**Competing interests:** The authors have declared that no competing interests exist.

already begun to reshape coral communities globally by reducing coral cover [8–12] and inducing shifts in benthic community composition toward more stress-tolerant coral taxa [13,14], sponges and soft corals [15], and fleshy seaweeds [16–18]. Without concerted action to address climate change, projections for the future of coral reefs are dire [2,19,20]. Still, certain coral populations have demonstrated capacity to persist in marginal environments or through intense environmental disturbances [21]. These corals are a source of hope for coral researchers and conservation practitioners alike, a sign that a future with functional coral reefs is still a possibility.

Disturbance events transform communities by inducing physiological stress that reduces the growth, survivorship, competitive ability, or fecundity of certain organisms [22–25]. For corals, stress can manifest as bleaching due to the breakdown of symbiosis between corals and their dinoflagellate symbionts in the family *symbiodiniaceae* [12]. Without their symbionts, corals lack their primary source of nutrition [26,27]. While mortality is not an inevitable outcome of coral bleaching [28–30], prolonged bleaching often results in partial or complete mortality of coral colonies. A variety of environmental stressors can cause coral bleaching, but bleaching is most commonly studied in relation to marine heatwaves [11,31]. In this context, bleaching onset is determined primarily by the intensity and duration of heat stress that corals experience [32,33]. Recent studies that document an increase in the magnitude of heat stress required to induce coral bleaching and mortality suggest that corals may possess the ability to acclimatize to heat stress over time [11,34,35].

Acclimatization refers to the process of phenotypic or epigenetic change that occurs within the lifespan of a single individual to cope with environmental stress [36]. In corals, acclimatization to heat stress can occur through a variety of mechanisms. For instance, Million et al. (2022) tracked *Acropora cervicornis* outplants through a bleaching event in Florida and found that genotypes with higher morphological plasticity had lower mortality and higher growth rates. Furthermore, these relationships strengthened over the course of the experiment, suggesting that acclimatization was actively occurring in response to heat stress [37]. In Kiribati, Claar et al. (2020) observed colonies of *Platygyra ryukyuensis* and *Favites pentagona* switch their symbiont communities from *Cladocopium* to *Durisdinium* during a bleaching event. This symbiont shuffling allowed colonies to regain pigmentation and recover from bleaching while heat stress remained elevated [38]. Heat-responsive genes have been found in coral tissues as well, and there is evidence that corals can modulate expression of these genes in response to heat stress [39,40]. In addition, the microbial communities associated with a coral "holobiont" can perform beneficial functions, such as nitrogen fixation and reactive oxygen scavenging, that have been shown to improve coral survivorship during heat stress [41].

Signs of coral acclimatization are widespread, having been documented in Australia [42,43], Hawaiʻi [44,45], Kenya [46], the Maldives [47], Malaysia [48], Palau [35], the Persian Gulf [49], and the Red Sea [50]. However, the cumulative stress of repeated bleaching is non negligible, and can reduce coral physiological performance over time [51,52] rather than induce acclimatization. Furthermore, when thermal stress is severe, bleaching-induced mortality can create a population bottleneck that reduces opportunities for acclimatization to occur [53,54]. Even following sequential heatwaves of moderate intensity, signs of acclimatization may not be evident at ecologically relevant scales.

At a global scale, there is evidence that coral bleaching thresholds have increased by 1°C since 1980 [55], with less severe bleaching observed on reefs that have been exposed to thermal stress previously [11,34]. While this increase in thermal tolerance could be due to widespread coral acclimatization, it could also be driven by other factors. Environmental conditions during a marine heatwave, such as high primary productivity, cloud cover, and turbidity, can act synergistically to reduce coral bleaching and mortality [12,56–58]. Furthermore, the

appearance of acclimatization on coral reefs could actually be the result of shifts in coral community composition toward thermally tolerant species due to bleaching-related mortality of vulnerable taxa [3,13,14,30] rather than improvements in the thermal tolerance of individual corals. As climate change intensifies, it is important that we understand whether changes in reef-scale thermal tolerance are driven by coral acclimatization or taxonomic turnover, or both.

Tracking the fate of individual corals through multiple marine heatwaves is one mechanism to better understand the drivers and prevalence of coral acclimatization to heat stress [25,59–61]. Many observational studies of coral bleaching use population-scale metrics to quantify bleaching trends (i.e., percent of colonies bleached [31]) rather than track the fate of individual coral colonies (but see [25,38,62,63]). Fate tracking can be hindered by the challenge of relocating and precisely delineating the boundaries of coral colonies *in situ*. This is particularly difficult for species where partial mortality causes colonies to split (fission) and regrow (fusion) over time [64,65]. This challenge can be ameliorated by large-area imaging technology (a.k.a. photogrammetry or Structure from Motion), which allows researchers to reconstruct 3D models of the reef benthos. These models can be used to precisely locate and track coral colonies through time [66,67]. Large-area imagery has been used quantify growth and mortality in demographically complex corals that display high rates of fusion and fission [61,68]. However, large-area imaging is a relatively recent technology in the coral reef field, which limits the temporal scope of most studies. Furthermore, any timeseries would need to document sequential marine heatwaves of similar magnitude in order to be useful for studying coral acclimatization, which can be a tall order.

Few studies have been conducted that 1) document the impact of sequential bleaching events on coral populations, 2) track individual coral colonies through time using large-area imaging, and 3) quantify patterns of coral bleaching, growth, and mortality in species with complex demographic trajectories involving fission and fusion dynamics (but see [69]). Here we track the fate of individual corals in leeward Maui to quantify the impact of successive bleaching events on coral growth and survivorship. Specifically, we sought to answer three questions. First, do individual corals, as well as coral populations, show evidence of acclimatization? Second, do corals with lower rates of bleaching outperform other corals over time? Finally, do different coral populations show similar responses to sequential heat stress in terms of bleaching, growth, and survivorship? This research holds particular relevance for coral restoration, given ongoing efforts to identify, cultivate, and outplant thermally tolerant coral genets as a form of climate-smart restoration [70]. If shifts in thermal tolerance in acclimatized corals are long lasting, or if certain locations or populations emerge as bastions of resilience, these corals could serve as parent colonies for restoration efforts [37,71].

## 2. Methods

### 2.1 Study sites

We surveyed populations of four coral taxa (*Montipora capitata*, *Montipora patula*, *Pocillopora* spp., and *Porites lobata*) at six sites in leeward Maui: Kahekili, Wahikuli, Olowalu, Ukumehame, Keawakapu, and Molokini. Our timeseries at these sites consists of five surveys (August 2014, November 2015, July 2017, October 2019, and June 2021), two of which coincide with documented bleaching events (2015 and 2019; Fig 1). These sites all represent hardbottom tropical coral reef habitat and range from 30.7% to 82.1% coral cover, 4.5 m to 10 m in depth, and 41 m to 487 m from shore (S1 Table in S1 File). The focal coral taxa studied here all represent common and important reef building species on Hawaiian coral reefs [72–74].

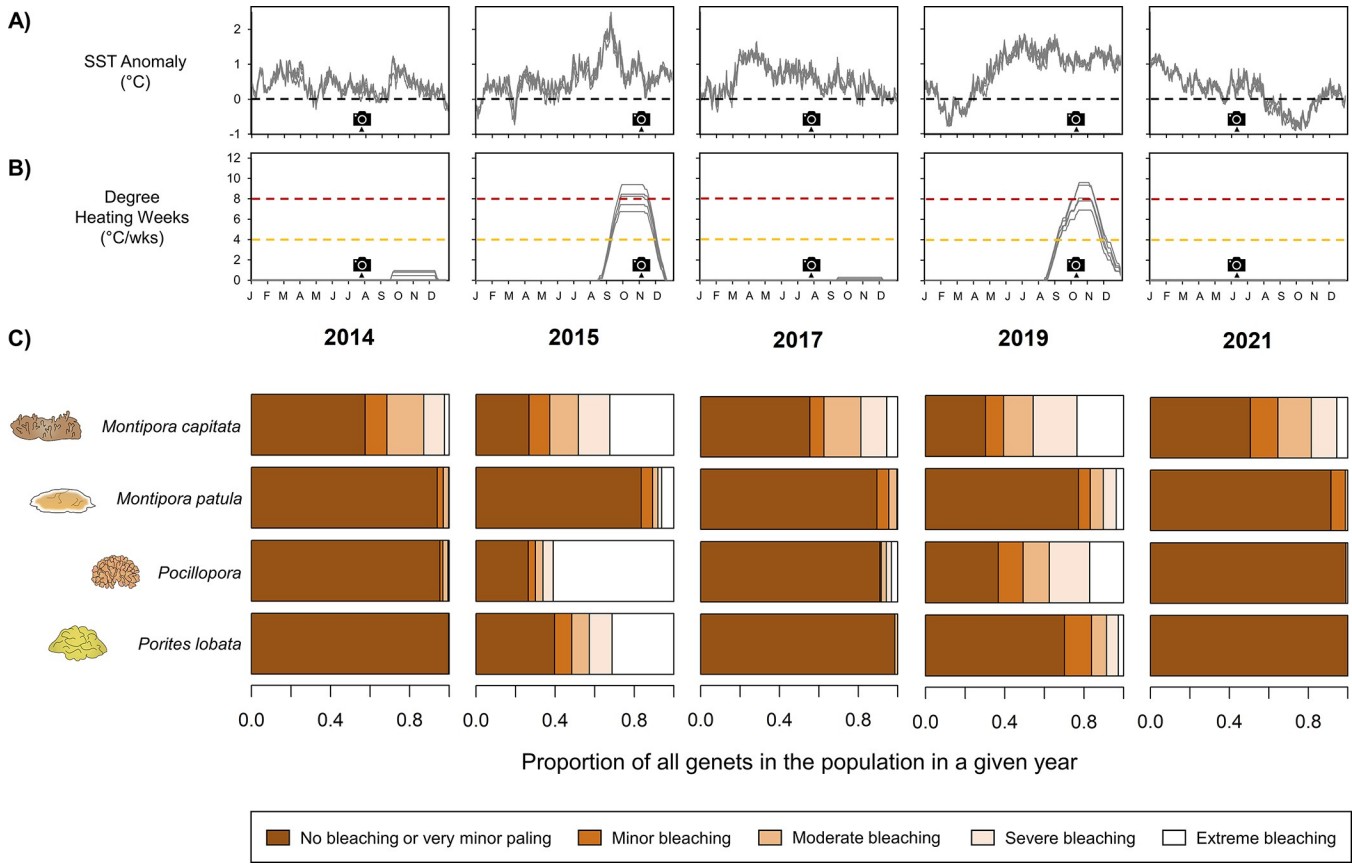

**Fig 1. Bleaching response of coral populations in each survey year.** Thermal stress is shown in each survey year for all sites in terms of (A) SST anomaly with respect to the 1985–2012 climatological mean, and (B) degree heating weeks (DHW). The black dotted line in (A) indicates a SST anomaly of 0. The yellow and red dotted lines in (B) denote the DHW threshold where bleaching and mortality, respectively, are expected to occur. Camera icons in (A-B) indicate the timing of large-area imagery surveys. (C) The bleaching response of each coral population is shown in each year. All genets are included, not just those that survived to the end of the timeseries.

The Hawaiian Islands have historically had a low incidence of bleaching compared to other coral reef ecoregions [75], but the archipelago was impacted by marine heatwaves in 2014, 2015, and 2019 that caused widespread coral bleaching [8,75–77]. While the spatial footprint and intensity of the 2015 and 2019 bleaching events varied throughout the Hawaiian archipelago, heat stress was similar in 2015 and 2019 for leeward Maui (mean increase of 0.42 degree heating weeks (DHW) from 2015 to 2019 across our sites, S1 and S2 Figs in S1 File, S1 Table in S1 File). This makes Maui an ideal natural experiment to identify signs of coral acclimatization, since we would expect similar levels of bleaching in 2015 and 2019 in response to similar levels of heat stress.

## 2.2 Data collection

We used large-area imaging (photogrammetry) to capture a 3D snapshot of coral reef condition at our six sites in leeward Maui in 2014, 2015, 2017, 2019, and 2021. Large-area imaging is the process of generating a composite visual reconstruction (i.e., 3D model, orthoprojection, DEM, etc.) via the overlap of many component images [66]. Large-area imaging has been used with increased frequency to archive coral reef structure and condition, and provides a basis for *in silico* field work to answer a variety of ecological questions [66,78–80]. Details of our large-

area imaging workflow are available elsewhere [67] and will be described in minimal detail here.

At each fixed site, divers entered the water with two D7000 SLR Nikon cameras (focal lengths of 18mm and 55mm) in Ikelite underwater housings. Divers marked the boundaries of each 10 x 10 m long-term monitoring plot with six calibration tiles, and placed four 0.5 m long scale bars within the plot. One diver swam approximately 1.5 m over the reef in a gridded pattern with the cameras, which were programmed to take a picture every second. This produced approximately 5,000 pictures per site. The diver imaged a core area of 10 x 10 m, and swam several meters beyond the calibration tiles to ensure that a buffer region (> 1 m wide) around the core area was also imaged.

Once imagery was collected, we used the software Agisoft Metashape (St. Petersburg, Russia) to build a dense point cloud, which we refer to here as a 3D model. After building the 3D models in Metashape, we loaded them into the custom software Viscore [81] for postprocessing. These postprocessing steps included 1) scaling the 3D model using the 0.5 m scale bars, 2) entering depth measurements collected at each calibration tile so that the 3D model could be oriented with respect to the sea surface, 3) manually aligning 3D models of the same site collected in different years, and 4) exporting a high-resolution top-down view of the model known as an orthoprojection. Each orthoprojection was 12 x 12 m and had a resolution of 1 mm per pixel. We used these orthoprojections rather than the 3D models for all subsequent data collection steps.

## 2.3 Coral tracing

Researchers used TagLab and ArcGIS Pro to trace patches of live coral tissue following the approach of Rodrguez et al. 2021 [68]. We found no effect of software on traced planar area (S3 Fig in S1 File), and a single annotator (OM) QCed all tracings in TagLab to control for any potential effect of annotator or software. We used the high-resolution imagery that underlies the 3D model (raw images) as a reference to assist with tracing and species ID. For this study, we chose to identify *Pocillopora* to the genus level because morphology is not a good indicator of species ID for *Pocillopora* in Hawai'i [82], whereas the three other taxa could be confidently identified to species. For *P. lobata*, we focused on massive and submassive growth forms to avoid confusion with the related branching coral *Porites compressa*.

After corals were traced, we used TagLab to "match" patches of live tissue that represented the same individual through time. This created a network of temporally linked coral patches (Fig 2). This enabled us to more accurately identify individual genets of corals such as *Porites* and *Montipora*, which readily exhibit fusion and fission over time due to partial morality [25,61,68,83]. Hereafter we use "patch" to denote a single contiguous region of live coral tissue in one timepoint, and "genet" to denote a network of patches interconnected through time (Fig 2). While we cannot definitively say that linked corals represent a single genetic individual (since we did no genetic testing), the precise tracking capabilities afforded by overlapping orthoprojections and associated raw images enable us to identify individual genets with a reasonable degree of confidence. Additionally, we visually inspected orthoprojections to identify potential instances of pseudoreplication arising from colony fission prior to our timeseries, and found minimal evidence to support this concern. We performed all subsequent analyses at the genet level, and considered a genet to have survived the full timeseries if at least one patch of that genet existed in both 2014 (our first sampling year) and 2021 (our final sampling year). For each genet, we calculated the total planar area ($cm^2$) in each timestep by summing the planar area of individual associated patches.

To achieve an appropriate sample size of *M. capitata*, *M. patula*, and *P. lobata*, we traced corals within 10 randomly placed non-overlapping 0.5 $m^2$ quadrats within the orthoprojection

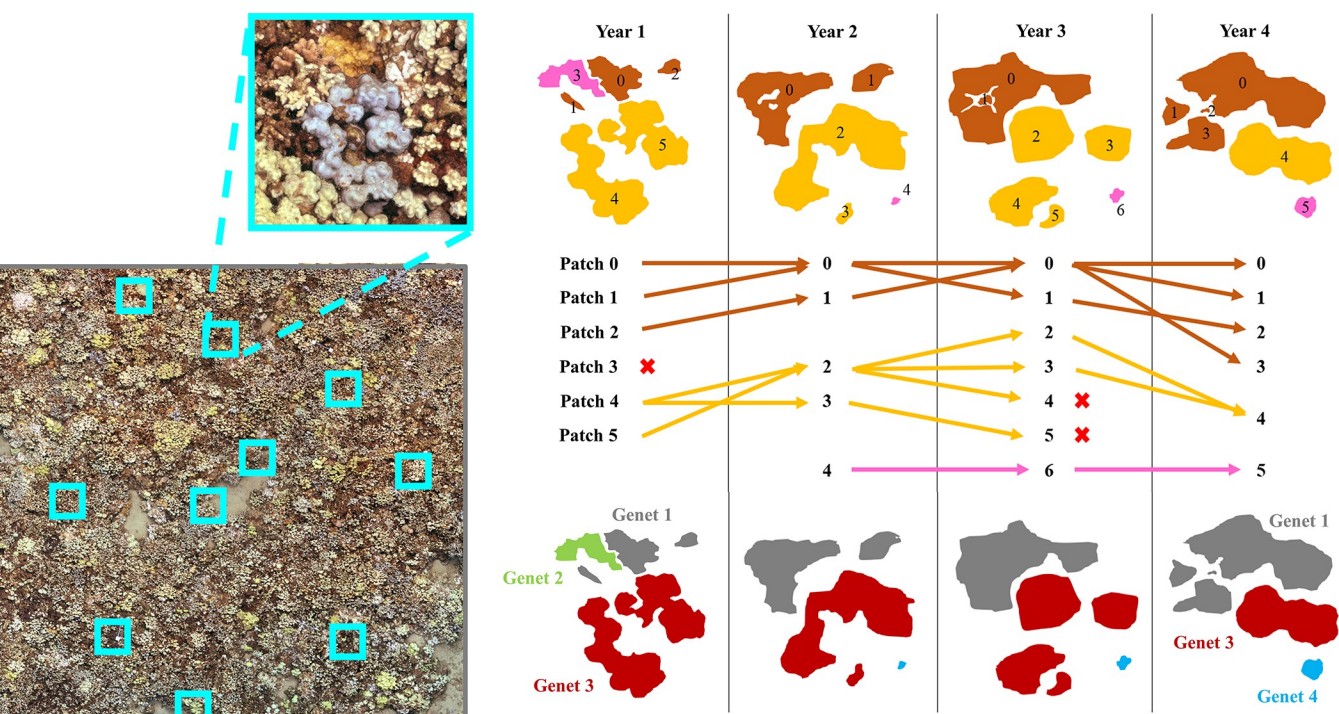

**Fig 2. Illustration of the coral tracing workflow using orthoprojections.** An example orthoprojection shows the location of randomly placed 0.5 m$^2$ quadrats (turquoise). For each quadrat, an annotator traced patches of live coral tissue > 5 cm in diameter. Patches were linked temporally to form a network of patches that were interconnected by fission (splitting via partial mortality) and fusion (growing together into a single patch). Annotators traced patches of live tissue < 5 cm in diameter and patches that fell outside of the quadrat if these patches were connected via fusion or fission to a genet within a quadrat. An example quadrat with traced patches is illustrated here, with fission and fusion represented by arrows. Patches are color coded by species and numbered sequentially in each year. Each linked network of patches is considered to be its own genet and assigned a unique ID number that is consistent across the timeseries. This is illustrated via color coding in the bottom panel. In this example, only genets 1 and 3 survived the full timeseries, while genet 2 experienced complete mortality between years 1 and 2. Genet 4 recruited to the reef between years 1 and 2, and as such wouldn't be included in this study, since our analysis focused on the cohort of coral genets that were alive at the start of the timeseries (genets 1, 2, and 3 in this example).

(Fig 2). We identified and traced all coral patches with a diameter > 5 cm whose centroid fell within the boundaries of a quadrat [68]. To account for fission and fusion dynamics, we also traced patches outside of a quadrat or < 5 cm in diameter if they were temporally linked to a genet inside a quadrat. At sites where < 40 genets were traced in our first timestep, we placed additional quadrats (up to 25 total) and continued tracing taxa until we reached 40 genets or until 25 quadrats had been placed. For *Pocillopora*, which was less abundant than other taxa, we identified and traced all patches within each 12 x 12 m orthoprojection. To ensure a balanced design for statistical analyses, we randomly selected 100 *P. lobata* genets for n = 2 sites where > 100 genets had been traced. The total number of coral genets traced per site, year, and taxa is shown in S2 Table in S1 File.

## 2.4 Bleaching

We assessed the extent and severity of bleaching for every patch of coral tissue > 1 cm$^2$ in each year of our timeseries. First, we visually estimated bleaching extent as the percent of a patch's area (0–100%) with some degree of paling unrelated to coral growth or disease. Focusing on this bleached tissue only, we then scored bleaching severity (from 0 to 3) based on the overall degree of paling observed (0 for practically no pigmentation loss; 1 for slight paling, 2 for significant loss of pigmentation, and 3 for almost or completely stark white; [68]). When

estimating both bleaching extent and severity, we used Viscore's Virtual Point Intercept inter-face [28] to reference the original imagery. This allowed us to view multiple angles of each coral patch to assess bleaching, enabling us to account for changes in tissue color due to light-ing or image quality.

Once we assessed bleaching extent and severity for all patches in all years, we calculated genet-level bleaching extent and severity in each timepoint by summing bleaching extent and severity across patches, weighted by patch area. Then, we incorporated bleaching extent and severity into a single bleaching metric for each genet in each timepoint (Fig 3). According to this metric, genets fell into one of five categories: no bleaching or very minor paling, minor bleaching, moderate bleaching, severe bleaching, or extreme bleaching. If a genet changed by more than one step between 2015 and 2019, the genet was considered to have an increased or decreased bleaching response over time. We considered genets that changed one bleaching step or less between 2015 and 2019 to have a stable bleaching response over time. Among these stable genets, those that exhibited moderate bleaching or higher were considered to have "high bleaching sus-ceptibility", and all other genets were considered to be "thermally tolerant". These bleaching responses (thermally tolerant, decreased bleaching response, increased bleaching response, and high bleaching susceptibility) were used as fixed factors to compare genets over time.

## 2.5 Statistical analysis

To test for signs of acclimatization among surviving genets in each population (corals of the same taxa within a given site), we employed a bootstrapping approach using only genets that were present in both bleaching events. We calculated the difference between each genet's bleaching score in 2015 and 2019, which produced an integer response variable ranging from -4 to 4, where 0 indicated no change in bleaching, -4 indicated extreme bleaching in 2015 and no bleaching in 2019, and 4 indicated no bleaching in 2015 and extreme bleaching in 2019. Then, we generated a null distribution of the median change in bleaching scores for each coral population via 100,000 resamples. P values were calculated as the proportion of all resamples where the population's median change in bleaching score was $\geq 0$, and we used $\alpha = 0.05$ as our threshold of significance. We did not test for acclimatization in two populations of *Pocillopora* spp. which had a low sample size of surviving genets ($< 10$ genets).

To test if coral populations at each site exhibited different bleaching responses, we created a contingency table of surviving genets based on their bleaching responses over time. We per-formed Fisher's exact test to test the null hypothesis that there was no relationship between site and bleaching response. We performed this test between all permutations of sites for each taxon and calculated adjusted p values using the Bonferroni correction for multiple compari-sons. We assigned post hoc letters at a significance level of $\alpha = 0.05$ to indicate significant dif-ferences in bleaching response between coral populations at different sites. We also pooled data by site and performed the same test to identify significant differences in bleaching response among coral taxa. Separately, we pooled observations across sites and ran a logistic regression of bleaching probability vs. genet size in 2015 and 2019 to quantify the relationship between bleaching and genet size. For this analysis, we coded bleaching as a binary variable (0 for genets with no signs of bleaching or minor bleaching, 1 for genets with moderate, severe, or extreme bleaching) in order to perform the logistic regression.

To test if acclimatized and thermally tolerant genets exhibited higher growth over the time-series than other corals, we performed an ANCOVA for each coral taxon, focusing on surviv-ing genets only. We used bleaching response (thermally tolerant, decreasing bleaching susceptibility, increasing bleaching susceptibility, and high bleaching susceptibility) as a fixed effect, only retaining those with a sample size $\geq 10$ for analysis. Our continuous predictor

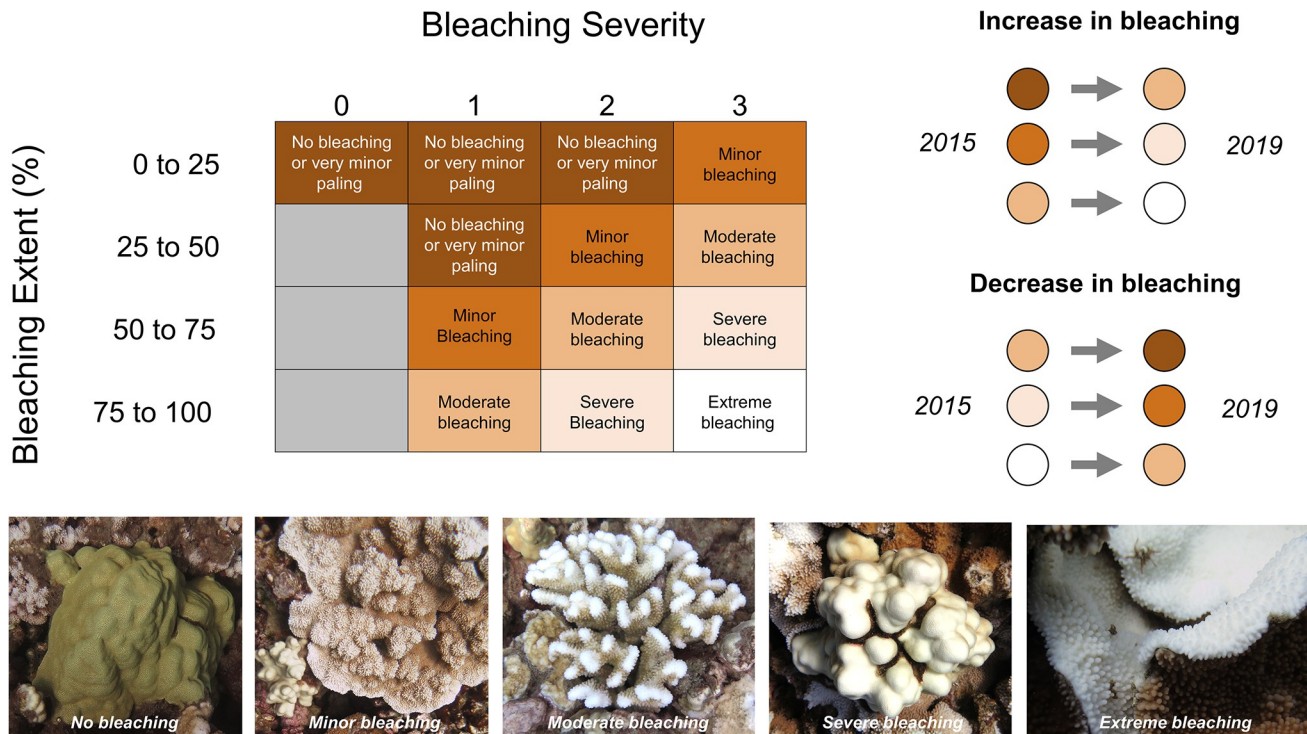

**Fig 3. Visual diagram of coral bleaching assessment criteria.** All patches of live coral tissue > 1 cm$^2$ were assigned a bleaching score based on their bleaching extent and severity. Each genet was assigned the most common bleaching score across all of its patches (weighted by patch area). The five bleaching scores were 1) no bleaching or very minor paling, 2) minor bleaching, 3) moderate bleaching, 4) severe bleaching, and 5) extreme bleaching. Representative pictures of each bleaching score are shown below. Genets were considered to have shown an increase or decrease in bleaching over time if they changed by two or more bleaching scores from 2015 to 2019.

variable was genet planar area at the start of the timeseries, and our response variable was genet planar area at the end of the timeseries. Genet planar area was natural log transformed for normality, and model assumptions were visually assessed using residual plots and quantitatively using Shapiro tests for normality and Cochran tests for homogeneity of variance. We performed a separate ANCOVA using site as a fixed effect to test for significant differences in coral growth between sites for each taxon. All models used a significance level of α = 0.05.

Finally, to test for differences in coral survivorship between sites, we performed a logistic regression using all traced genets (not just survivors) for each taxon. We used site as a fixed effect, log transformed planar area of genets in 2014 as our predictor variable, and coded genet survivorship as a binary response variable (0 = genet did not survive until 2021, 1 = genet survived from 2014 to 2021). There were significant interactions between multiple sites, so to facilitate interpretation we conducted pairwise comparisons between sites that didn't interact. We calculated adjusted p values using the Bonferroni correction for multiple comparisons at a significance level of α = 0.05. To test for differences in coral survivorship between taxa, we also pooled genets by site and ran a logistic regression of survivorship vs. initial genet size. We completed statistical analyses using the 'dplyr', 'emmeans', 'multcomp', 'coin', and 'rstatix' packages in R v. 4.0.5 [84–89].

## 3. Results

In total, we tracked the fate of 1,832 coral genets across six sites from 2014 to 2021. *Pocillopora* exhibited exceptionally low survivorship: 15.7% of *Pocillopora* genets survived from 2014 to

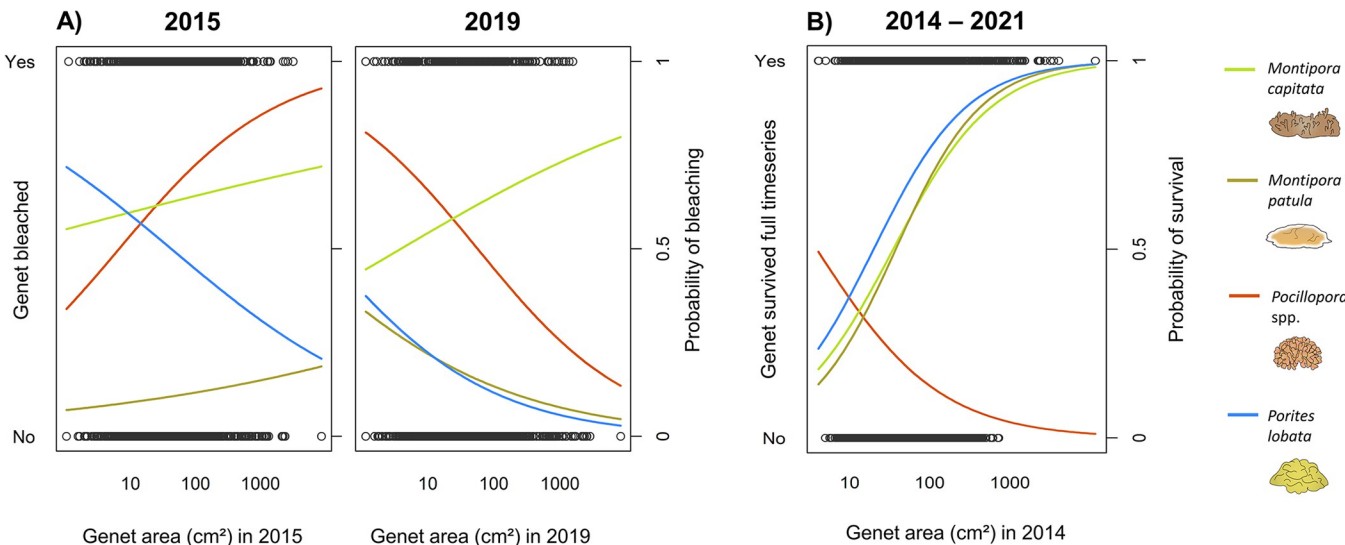

**Fig 4. Bleaching and survival probabilities as a function of genet planar area.** (A) The probability of bleaching is shown for each taxon in 2015 and 2019. Bleaching was converted into a binary variable for this analysis (1 for genets with moderate, severe, or extreme bleaching, 0 for genets with minor or no bleaching). (B) The probability that a genet survived to the end of the timeseries (2021) is shown as a function of its planar area at the start of the timeseries (2014). Planar area was a significant predictor of survivorship for all taxa (p < 0.001 for all taxa), and was positively related to survivorship for *Montipora* and *Porites*, but negatively related to survivorship in *Pocillopora*.

2021, compared to 54.4% of *M. capitata* genets, 51.3% of *M. patula* genets, and 63.2% of *P. lobata* genets. Bleaching prevalence was high for *Pocillopora* and *M. capitata* during both bleaching events: 70.0% of *Pocillopora* and 62.0% of *M. capitata* genets that survived to 2015 experienced moderate, severe, or extreme bleaching in the first bleaching event, while 50.4% of *Pocillopora* and 60.3% of *M. capitata* genets that survived until 2019 experienced moderate, severe, or extreme bleaching in the second bleaching event (Fig 1C). This contrasts with *M. patula*, which experienced low bleaching during both bleaching events (10.6% in 2015, 16.8% in 2019), and *P. lobata*, which experienced high bleaching in 2015 (51.0%) but low bleaching in 2019 (16.0%; Fig 1). We also observed an inconsistent relationship between genet size and the probability of bleaching across taxa and bleaching events (Fig 4). In 2015, the probability of bleaching increased with genet size for *Pocillopora* (p < 0.001) but decreased with genet size for *P. lobata* (p < 0.001), and exhibited no significant relationship with size for either species of *Montipora* (*M. capitata* p = 0.348; *M. patula* p = 0.364). In 2019, bleaching probability increased with genet size for *M. capitata* (p = 0.047) but decreased with genet size for the other taxa (*M. patula* p = 0.027; *Pocillopora* p = 0.021; *P. lobata* p = 0.002).

### 3.1 Evidence of acclimatization

Patterns of acclimatization over the course of this study differed between coral taxa. We found no evidence of acclimatization at the population level for either species of *Montipora* (Fig 5A and 5B), with more variability in bleaching over time in *M. capitata* compared to *M. patula*. Conversely, we did find evidence for acclimatization in specific *Pocillopora* and *P. lobata* populations (Fig 5C and 5D): *Pocillopora* from Kahekili and Olowalu exhibited less severe bleaching in 2019 compared to 2015 (p < 0.001 and p = 0.042, respectively). *Pocillopora* that survived until 2019 from Ukumehame demonstrated increasing bleaching susceptibility over time, although the sample size for that population was small. For *P. lobata*, we found significant evidence of population-scale acclimatization at Keawakapu (p = 0.023) and Molokini (p = 0.010). Across all sites, *P. lobata* genets were most likely to exhibit signs of acclimatization, ranging

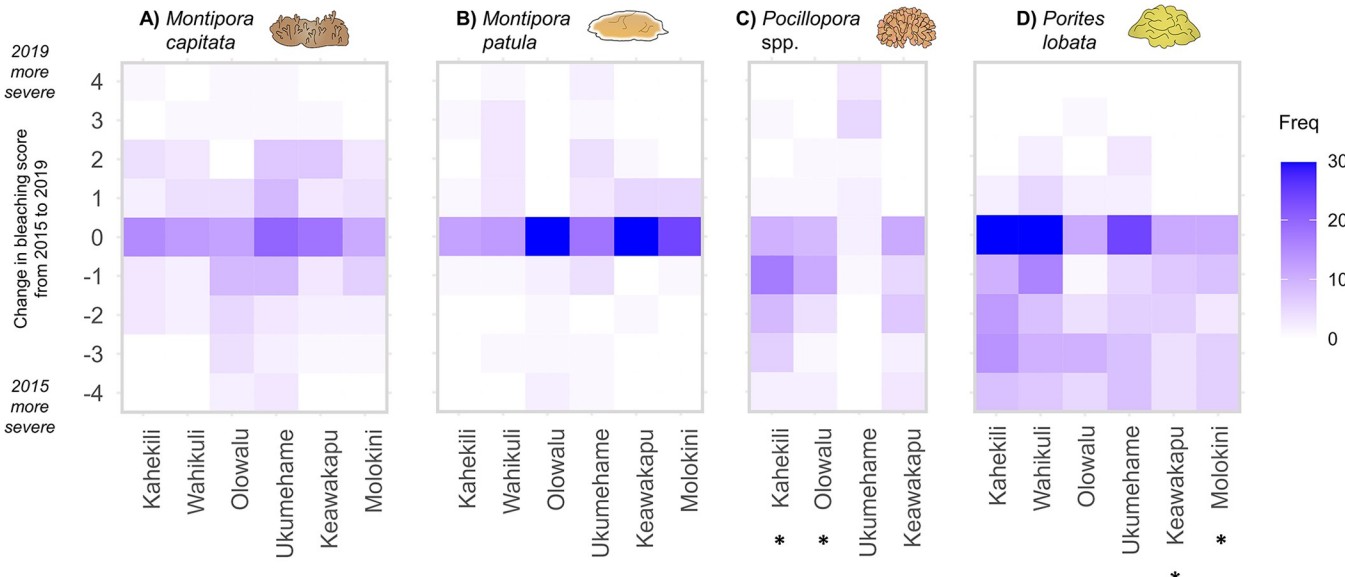

**Fig 5. A heatmap illustrating how genet bleaching scores changed between 2015 and 2019.** For each genet that survived through 2019, we calculated the difference in bleaching scores between 2015 and 2019. Negative values indicate more severe bleaching in 2015, positive values indicate more severe bleaching in 2019, and 0 indicates no change in bleaching. We observed significant decreases in bleaching over time for *Pocillopora* populations at Kahekili (p < 0.001) and Olowalu (p = 0.042), and for *Porites lobata* populations at Keawakapu (p = 0.023) and Molokini (p = 0.010). *Pocillopora* at Molokini and Wahikuli were excluded from the analysis, since the vast majority of genets at those sites perished after the first bleaching event in 2015.

from a low of 18.0% of all *P. lobata* genets at Wahikuli to 39.1% at Olowalu. It was more common, however, for *P. lobata* genets to exhibit the same bleaching severity in both years (often no bleaching), particularly at Kahekili, Wahikuli, and Ukumehame.

### 3.2 Population bleaching response

We tracked genets based on their response to thermal stress in 2015 and 2019, and found that the bleaching response of genets strongly differed between species (p < 0.001). In general, *M. capitata* had the largest proportion of genets that were highly susceptible to bleaching, *M. patula* had the largest proportion of thermally tolerant genets, and *P. lobata* had the largest proportion of genets that exhibited signs of acclimatization (Fig 6). However, we observed site-level variation in these trends, with some populations exhibiting higher proportions of thermally tolerant and acclimatized genets than others. Most strikingly, the bleaching response of *M. capitata* genets at Kahekili was significantly different from all other sites except for Keawakapu, driven by a higher proportion of thermally tolerant individuals (Fig 6). We also observed a larger proportion of *P. lobata* genets that were highly susceptible to bleaching at Wahikuli compared to Olowalu, which had more genets exhibiting signs of acclimatization. We also found significantly more *M. patula* genets with increasing bleaching susceptibility at Wahikuli and Ukumehame compared to Olowalu (Fig 6). We did not observe any significant differences in bleaching response between *Pocillopora* populations, although our analysis was limited to genets from Kahekili, Olowalu, and Keawakapu due to the small sample size of surviving genets at our other three sites (Fig 6).

### 3.3 Growth

For genets of *M. capitata* and *Pocillopora* that survived the full timeseries, we found no evidence that genet growth or shrinkage was a function of a genet's bleaching response over time (p = 0.052

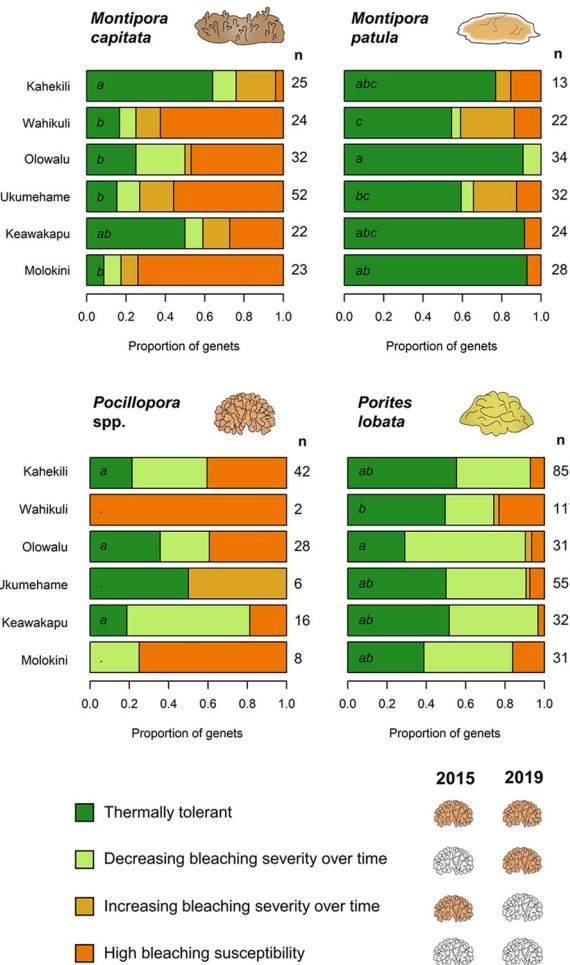

**Fig 6. Surviving genets shown by their response to heat stress in 2015 and 2019.** Dark green indicates genets that didn't bleach in either event, light green indicates genets that bleached more in 2015 than 2019, yellow indicates genets that bleached more in 2019 than 2015, and orange indicates genets that bleached in both events. Letters denote significant differences between populations in terms of their bleaching response. The number of surviving genets in each population is shown to the right of each plot.

and p = 0.073 respectively; Fig 7A). We chose not to conduct an ANCOVA for *M. patula* since genets exhibited minimal variation in bleaching response (78% of genets didn't bleach in either event), although we observed no obvious relationship between bleaching response and growth or shrinkage for this species either (Fig 7A). Interpreting the results for *P. lobata* was more complex due to the presence of a significant interaction between initial genet size in 2014 and final genet size in 2021 ($p_{site}$ = 0.004, $p_{interaction}$ = 0.003). Small *P. lobata* genets ($< 100$ cm$^2$ in 2014) that survived the full timeseries didn't exhibit a relationship between bleaching history and genet growth or shrinkage. However, larger genets ($> 100$ cm$^2$ in 2014) that experienced repeated bleaching were more likely to experience partial mortality and genet shrinkage compared to thermally tolerant individuals, which exhibited a nearly 1:1 relationship between planar area in 2014 and 2021. Thus, bleaching history appears to have more of an impact on the growth trajectory of surviving genets in *P. lobata* compared to the other taxa studied here.

While bleaching history didn't impact coral growth for three of the four taxa studied here, we observed site-level differences in growth for all focal taxa (Fig 7B). In particular, *M. patula* and *Pocillopora* from Keawakapu and *P. lobata* from Molokini exhibited high rates of partial

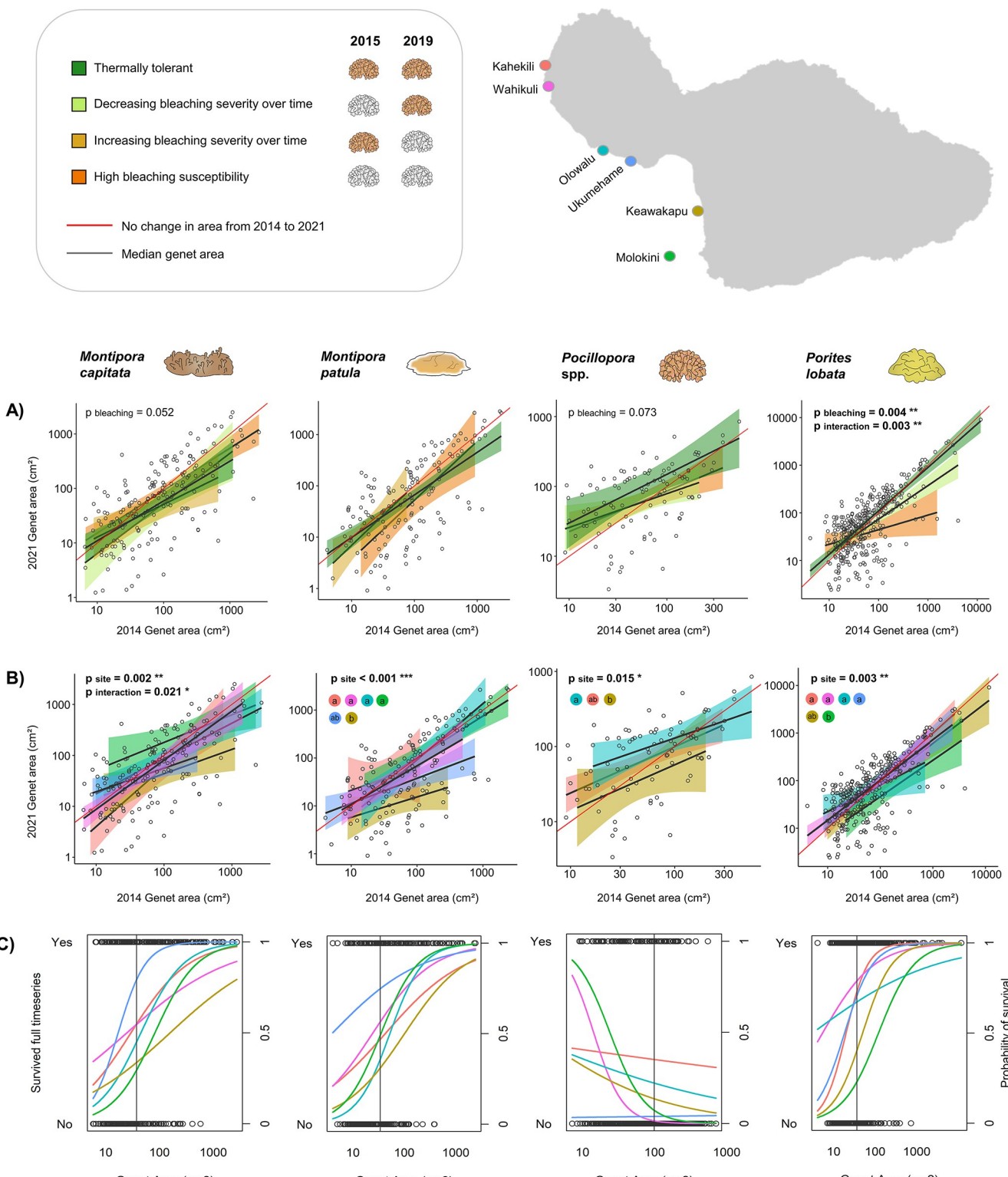

**Fig 7. Patterns of growth and survivorship by site and bleaching history.** (A-B) Log-scale plots show patterns of growth and shrinkage in genets that survived the full timeseries as a function of (A) genet bleaching history and (B) site. Genets that fall on the red diagonal line experienced no change in planar area between 2014 and 2021, those above it experienced growth, and those below it experienced shrinkage. (C) For each taxon, site-level patterns of survivorship to the end of the timeseries are shown as a function of genet planar area in 2014. The vertical gray line in each plot represents the median planar area of that taxon in 2014. The key in the top left corresponds to bleaching histories in (A). Sites on the map denote color coding in (B-C). Letters in (B) denote significant differences between sites.

mortality and shrinkage over the course of the timeseries. For *M. capitata*, smaller genets from Molokini ($< 100$ cm$^2$ in 2014) appear to have grown more than genets from other sites, but this difference disappears for larger genets. Instead, for larger *M. capitata* genets ($> 100$ cm$^2$ in 2014), Keawakapu appears to have significantly higher rates of partial mortality and shrinkage.

### 3.4 Survivorship

We found a significant relationship between survivorship from 2014 to 2021 and initial genet planar area for all taxa ($p < 0.001$). *M. capitata*, *M. patula*, and *P. lobata* exhibited a strong positive relationship between 2014 planar area and probability of genet survival through 2021, with no significant differences in survivorship between those three taxa. Conversely, *Pocillopora* genets exhibited a strong negative relationship between 2014 planar area and survivorship. The probability of survivorship was considerably lower for *Pocillopora* than for the other taxa. For example, a *Pocillopora* genet with a planar area of 100 cm$^2$ in 2014 would be expected to survive until 2021 just 13.8% of the time, compared to a 67.6% probability of survivorship for *M. capitata*, 68.8% for *M. patula*, and 76.7% for P. *lobata* (all for genets with a planar area of 100 cm$^2$ in 2014). The vast majority of *Pocillopora* mortality occurred between 2015 and 2017, likely due to the 2015 bleaching event, while mortality for the other three taxa was more evenly distributed across the timeseries (S4 Fig in S1 File).

As with coral growth, we observed site-level differences in genet survivorship from 2014 to 2021. We identified sites with the highest and lowest survivorship through pairwise comparisons of sites that did not exhibit significant interactions (Fig 7C; S3 Table in S1 File). Ukumehame stood out for high survivorship of *M. capitata*, *M. patula*, and *P. lobata*, Kahekili exhibited high survivorship of both *Pocillopora* and *P. lobata*, and Wahikuli exhibited high survivorship of *P. lobata*. Conversely, Keawakapu had low survivorship of *M. capitata*, *M. patula*, and *P. lobata*, and Molokini had low survivorship of *M. capitata*, *Pocillopora*, and *P. lobata*. Olowalu had low survivorship of *M. capitata* compared to other sites, while Ukumehame and Wahikuli both had low survivorship of *Pocillopora*.

## 4. Discussion

We tracked the fate of hundreds of corals through sequential marine heatwaves of moderate intensity. By comparing the bleaching response of individuals through time, we sought to document evidence of acclimatization at organismal and population scales. Taken in concert with growth and survivorship data, we were able to test whether past bleaching responses influenced coral growth for corals that survived the full timeseries. Finally, variation in coral bleaching, growth, and survivorship among six sites in leeward Maui reflected the unique environmental history of each reef. Interactions between environmental and anthropogenic factors likely contributed to site-level variation in coral performance over time.

### 4.1 Acclimatization

We did not find evidence of widespread acclimatization on reefs in leeward Maui, although a few populations did exhibit significant reductions in bleaching severity and extent over time, most notably *Pocillopora* at Kahekili. While signs of acclimatization at the population level were limited, individual genets (primarily *P. lobata*) did show decreased bleaching severity and extent over time. Massive *Porites* employs a stress tolerant life history strategy [25,61,90,91] and has a demonstrated ability to acclimatize to repeated heat stress [43,57,92]. Massive *Porites* taxa are generally among the most thermally tolerant corals on Indo-Pacific reefs [14,31], although thermal tolerance relative to other taxa has been shown to decline under severe thermal stress [54]. The heat tolerance of *Porites* contrasts with that of faster

growing coral taxa such as *Acropora* and *Pocillopora*, which have been consistently found to be among the least resistant to thermal stress [14,45,54,76,93]. These taxa tend to experience more mortality but have demonstrated the capacity to rapidly recolonize reef communities after heat stress abates [13,25,29]. However, severe mortality events can hamper recovery by eliminating sources of recruitment [53]. This could be a factor at play in Maui, where *Pocillopora* abundance has remained consistently low following widespread bleaching and mortality in 2015.

Compared to *Porites*, evidence for acclimatization in *Montipora* and *Pocillopora* is more mixed. Coles et al. (2018) found that *M. capitata* and *Pocillopora damicornis* from Kāneʻohe Bay took longer to bleach and showed higher survivorship when experimentally reared at 31˚C in 2017 compared to in 1970 [36]. Association with thermally-tolerant symbionts has also been found to increase bleaching resistance in *M. capitata* [94–96] and *Pocillopora* [97], although studies have found mixed evidence for symbiont shuffling as a mechanism to achieve long term acclimatization in *M. capitata* [95,98]. Rather, it appears that high phenotypic variation in pigmentation exists for *M. capitata*, with both pale and highly pigmented variants observed on reefs even in the absence of heat stress (Fig 1C; [60]). While studies have observed a relationship between pigmentation and performance under heat stress in *Montipora* in Hawaiʻi [74,94], we find little evidence that more pigmented variants are being selected for and becoming more dominant over time. Indeed, we found considerable phenotypic variation in *M. capitata* bleaching, including shifts in the pigmentation of individual genets over time (both darkening and paling) during both bleaching and non-bleaching years (Figs 1C and 5A). Yet we observed no reductions in *M. capitata* bleaching at a population scale over our time-series. These results likely reflect *M. capitata*'s ability to use heterotrophy to supplement nutritional deficits in the absence of symbionts [26,92], which could reduce the need for *M. capitata* to undergo acclimatization through some other mechanism. While *Pocillopora* survivors showed patterns of reduced bleaching consistent with acclimatization at certain sites, we are unable to say what mechanism is responsible for those shifts. It is possible that cryptic species diversity could explain site-level differences in bleaching, given that we were only able to identify *Pocillopora* genets to the genus level.

Compared to extreme heatwaves that cause widespread mortality, moderate thermal stress would be expected to produce more variable bleaching responses among individual corals and promote natural selection for thermal tolerance [28,33,54,99]. Both the 2015 and 2019 heat stress events were moderate in severity and duration in Hawaiʻi compared to elsewhere in the Pacific [9,10,28], just barely surpassing the 8 DHW threshold where coral mortality is be expected. The considerable mortality we observed from 2014 to 2021 indicates that certain individuals were being selected against, whereas survivors showed a lack of cumulative stress from bleaching, as evidenced by the absence of any relationship between growth rate and bleaching history. Selection for thermal tolerance has been observed on reefs elsewhere [3,13,14,34], resulting in notable shifts in coral community composition toward stress-tolerant species. Admittedly, it is also possible that the timing of our 2019 survey influenced our ability to detect acclimatization. We collected imagery just before peak heat stress was reached in October, so it is possible that bleaching could have continued to worsen for several weeks after our imagery was collected. That being said, our results are consistent with other analyses that have found less severe bleaching in 2019 compared to 2015 in Hawaiʻi [77,100], so it is unlikely that the timing of our surveys alone can explain the results we observe here.

## 4.2 Growth

While the bleaching responses we observed are consistent with known taxonomic patterns of thermal tolerance in Hawaiian corals [45,72,74,77,92], we found a striking disconnect between

bleaching and growth over the course of our timeseries. With the exception of large *Porites*, genets that repeatedly bleached and survived did not demonstrate slower growth or more partial mortality than thermally tolerant or acclimatized genets over the course of our timeseries. The lack of relationship between bleaching and coral growth that we observe could be the result of complex physiological trade-offs between maintenance and growth. For instance, *Pocillopora* colonies in Guam that were experimentally acclimated to 31˚C prioritized lipid accumulation and tissue growth over skeletal extension [101]. More generally, studies have found that energetic costs associated with hosting the heat-tolerant symbiont *Durusdinium* can result in trade-offs for the coral host including reduced growth and fecundity [98,102]. However, other studies of heat stressed corals fail to document trade-offs between thermal tolerance and growth at ecological scales [103,104]. Given this mixed evidence, even among studies of the same species, it seems likely that trade-offs between heat tolerance and growth are site specific and dependent on local environmental conditions and exposure history [104].

Another explanation for the decoupling we observe between bleaching responses and coral growth could be the complexity of fusion and fission dynamics in our focal taxa. Partial mortality can cause a coral genet to split into multiple independent fragments, or ramets, which can either regrow and fuse back into a single genet, or continue living as functionally separate individuals. These individual ramets each experience their own trajectory of growth, shrinkage, and mortality based on interspecific interactions and hyper-localized environmental conditions [64,65,105,106]. These processes make it more complicated to identify and track the fate of "individuals" and can result in complex growth trajectories across an entire genet. Partial mortality, fusion, and fission are especially apparent in encrusting and massive corals like *M. capitata*, *M. patula*, and *P. lobata* [60,83], and can help these individuals avoid whole-colony mortality following environmental stress [107] even if most tissue is lost. For example, partial mortality may be concentrated on the top of coral colonies following exposure to higher irradiance during a bleaching event [74]. However, these corals can persist via regrowth of tissue that survived bleaching in shaded or cryptic regions of the colony [60]. Due to these complexities, most studies tracking coral growth and survivorship focus on taxa with identifiable, discrete colonies that display little fission or fusion, such as *Pocillopora* [108,109]. Our results demonstrate that such approaches may be inadequate for describing demographic and recovery dynamics in locations like Hawai'i, where fusion and fission predominate.

## 4.3 Survivorship

While bleaching and long-term growth appear largely decoupled on Maui's reefs, genet size was a significant predictor of both bleaching and survivorship for multiple taxa. Small genets of *P. lobata* and *M. patula* were more likely to bleach and less likely to survive the full duration of our timeseries, which is consistent with literature on size-based survivorship in corals. Small coral colonies have a greater surface area to volume ratio and less energy reserves, which make them more vulnerable to whole-colony mortality, whereas injury in larger corals is more likely to lead to partial mortality [64,108,110]. The influence of coral size on bleaching is less straightforward. Some studies have found small colonies to be more susceptible to bleaching [111], while others have found large colonies to be more susceptible [54,112], and still others document no relationship between bleaching and colony size [75,113]. Interestingly, while we observed divergent bleaching responses for *M. capitata* and *M. patula*, both in severity and as a function of size, these species displayed nearly identical patterns of survivorship. This supports other studies in Hawai'i that have found a disconnect between bleaching and survivorship in *M. capitata* [45,60] and likely reflects taxonomic and life history similarities between *Montipora* species.

In *Pocillopora*, size-dependent patterns of bleaching and survivorship reversed over time, with more bleaching and mortality in larger genets in 2015, followed by more bleaching and mortality in smaller genets in 2019 (S5 Fig in S1 File). While we would expect to see the highest mortality in smaller size classes in *Pocillopora* [61,108], higher mortality in large *Pocillopora* colonies in response to bleaching has been observed before [93,109]. In Moorea, the coexistence of multiple cryptic species of *Pocillopora* led to the appearance of a positive relationship between colony size and mortality after a bleaching event [114]. However, through genetic testing, Burgess et al. (2021) revealed that the largest *Pocillopora* colonies all belonged to a single haplotype with high bleaching susceptibility. Thus, the shift in size-dependent mortality we observe from 2015 to 2019 could represent the loss of cryptic species in *Pocillopora* in Maui, although genetic testing would be needed to confirm this. If a loss of cryptic diversity is responsible for the survivorship trends we observe here, it would suggest that bleaching events are driving a shift in coral community composition in Maui toward taxa with higher thermal tolerance.

## 4.4 Variation between sites

The severity of thermal stress was remarkably consistent across study sites in leeward Maui and between bleaching events, although local scale variability undoubtedly exists that is not captured in satellite data at the 5km grid scale [115]. Based on the satellite DHW data available, we would expect similar levels of bleaching and mortality across our sites and through time, with slightly lower levels of mortality expected in northern sites compared to southern sites (S1 Table in S1 File), assuming a static threshold of 8 DHW for mortality [8,99,116]. Importantly, despite their shared thermal history, these sites each experience a unique combination of local environmental and anthropogenic impacts that could also explain divergent patterns of coral bleaching, growth, and recovery across populations [72,83,117,118]. Indeed, chronic stressors such as urban runoff, sedimentation, coral disease, overfishing of herbivores, and algal overgrowth have been shown to be more important contributors to coral tissue loss in Maui than acute pulse events [107,119].

In our study, coral populations at Kahekili and Olowalu showed the greatest resilience to thermal stress, while populations at Keawakapu and Molokini showed the least resilience (S4 Table in S1 File). Corals at Kahekili in particular showed higher rates of thermal tolerance (*M. capitata*) and acclimatization (*Pocillopora*) than conspecifics elsewhere, even compared to nearby populations at Wahikuli that experienced near identical heat stress. Kahekili has a complex history of management and anthropogenic impacts. Situated just offshore of the Lahaina Wastewater Reclamation Facility, Kahekili's reefs have been subject to elevated concentrations of nitrogen from wastewater effluent that seeps out of nearby injection wells [117,120]. Elevated nutrients at Kahekili have contributed to occasional mass algal blooms that overgrow corals [118–120], which prompted the state Division of Aquatic Resources to establish the Kahekili Herbivore Fisheries Management Area in 2009 [121]. Efficacy of these protections has fluctuated over time, but there is evidence that protecting herbivorous fish has reduced macroalgae and increased coverage of crustose coralline algae [121], a less effective competitor [122] that promotes coral recruitment [123].

A combination of fisheries management and elevated nutrients could explain the improved bleaching response we observed at Kahekili. Moderate nutrient enrichment has been shown to delay the onset of bleaching [124] and reduce bleaching-associated mortality by improving the photosynthetic efficiency of symbionts [30] or enhancing water column chlorophyll-a, which in turn can reduce irradiance and support coral heterotrophy [125]. However, it is important to note that the interactive effects of nutrient enrichment and macroalgae have been found to

be especially harmful for corals during and immediately after bleaching events [126,127], since nutrients enhance the growth and competitive ability of macroalgae [122]. This suggests that control of macroalgae through fisheries management is fundamental to the bleaching resistance that we document here.

It is possible that repeated algal blooms over multiple decades could have already selected for stress-tolerant corals at Kahekili, enhancing community resistance to disturbance [77,128]. This pattern has been observed on human-impacted reefs elsewhere in Hawai'i [129]. The reverse of this phenomenon could explain the poor performance of corals at Molokini, a highly protected site located offshore of Maui within the Molokini Shoal Marine Life Conservation District. It is possible that the protection and remoteness of Molokini allowed more sensitive coral genets to persist until 2015, which would explain the lower survivorship compared to more heavily impacted sites such as Kahekili. However, we also document low survivorship at Keawakapu, a site with a similar suite of anthropogenic impacts to Kahekili [117,130], albeit without fisheries management. The small number of sites considered here, combined with each site's multifaceted and unique history, complicate any assumptions of causality. While the exact sources of variation in coral performance are uncertain, it's clear that that local anthropogenic impacts are interacting with bleaching history and species composition to shape the trajectory of each site's benthic community over time.

## 4.5 Implications for coral restoration

For coral restoration to be effective, outplanted corals must be able to persist through future climate-related disturbance events [80,131,132]. To this end, restoration practitioners have sought to propagate thermally tolerant phenotypes or genotypes [70,71] in hopes that these outplants prove more resilient over time. Here, we show that thermally tolerant genets do not always outperform their bleached conspecifics, whether due to trade-offs associated with thermal tolerance, complex fusion and fission growth pathways, or the confounding effects of local environmental factors. For example, we found that sites with a higher proportion of tolerant individuals were not necessarily sites with the highest survivorship (S4 Table in S1 File).

This disconnect holds important implications for coral restoration efforts, both in Hawai'i and elsewhere. Our findings demonstrate how sequential bleaching events shape coral communities, weeding out susceptible individuals and leaving behind a population with the demonstrated ability to survive thermal stress. These survivors now constitute the bulk of Maui's reef communities and could act as a source population for coral restoration efforts. We do not find support for targeting specific phenotypes of *Montipora* or *Porites* based on perceived thermal resistance, since we observed growth and survivorship across a spectrum of phenotypes. The best available indicator of survivorship under future thermal stress is survivorship through past thermal stress events, which can be assumed for large colonies present on Maui's reefs today.

The one exception to this conclusion is *Pocillopora*. Given the low thermal tolerance and low survivorship we observed here, any restoration projects focused on *Pocillopora* should seek to identify and raise thermally tolerant strains or individuals, as they would have the best potential to survive future heat stress. However, based on survivorship rates observed here, it is likely that even thermally tolerant outplants of *Pocillopora* would have lower survivorship than outplants of more stress-tolerant taxa, such as *Porites*. Ultimately, the decision of which taxa to select for coral restoration depends on the ecological and social goals of the project [131,132]. For instance, *Pocillopora* restoration could be used as a mechanism to boost recruitment or diversity on reefs where *Pocillopora* abundance has remained depressed following repeat bleaching. If the goal of restoration is simply to increase coral cover and persistence through future heat stress, *Porites* and *Montipora* are more advisable options.

## 5. Conclusions

Over seven years and two bleaching events of similar magnitude, coral populations in Maui demonstrated hallmarks of selection and resilience. While we did not find widespread support for reef-scale acclimatization, we did find evidence of possible acclimatization for individual genets, primarily in the massive coral *P. lobata*. We also document selection against thermally susceptible individuals, with implications for the overall diversity and future thermal tolerance of Maui's reefs. One important implication of selection is that bleaching and growth were largely decoupled for corals that survived repeated exposure to heat stress in Maui. The resilience of these survivors will no doubt be tested by future heatwaves. In the meantime, it will be important to the track colony-level patterns of bleaching, growth, and survivorship for multiple taxa (including those with fusion and fission dynamics) at broad ecological scales to understand geographic trends in coral acclimatization and selection. These efforts should harness advances in large-area imaging and coral genetics to identify the underlying mechanisms responsible for acclimatization and the impact of selection on coral populations. This knowledge in turn can be used to improve projections of reef persistence under climate change, and inform decisions related to coral restoration.

## Supporting information

**S1 File. Contains S1-S5 Figs and S1-S4 Tables.**
(DOCX)

## Acknowledgments

We would like to thank the divers who helped conduct the large-area imagery surveys, including Anela Akiona, Donna Brown, Samantha Clements, Emily Kelly, Susan Kram, Travis Matteson, Melissa Torres, and Darla White. Special thanks to Emily Kelly for establishing this large-area imagery timeseries back in 2014. Field work would not have been possible without the support of local partners in Maui, including Ultimate Whale Watch, Dive Maui, Maui Divers, Lee James, Peter and Toni Colombo, Craig and Amy Venema, Don McLeish, Will and Megan Dailer, and George and Donna Brown. We would also like to thank undergraduate research volunteers Solomon Chang, Andre Lai, Veronika Pearson, Varun Sachin Shirhatti, and Victoria Vasquez for their tireless work tracing coral patches. We also thank Tom Oliver, Courtney Couch, and Mary Donovan, who provided input on project conception and initial results, and Adi Khen for designing the coral graphics used in our figures.

## Author Contributions

**Conceptualization:** Orion S. McCarthy, Morgan Winston Pomeroy, Jennifer E. Smith.

**Data curation:** Orion S. McCarthy, Morgan Winston Pomeroy.

**Formal analysis:** Orion S. McCarthy, Morgan Winston Pomeroy.

**Funding acquisition:** Orion S. McCarthy, Jennifer E. Smith.

**Investigation:** Orion S. McCarthy.

**Methodology:** Orion S. McCarthy, Morgan Winston Pomeroy.

**Project administration:** Orion S. McCarthy.

**Resources:** Jennifer E. Smith.

**Software:** Orion S. McCarthy, Morgan Winston Pomeroy.

**Validation:** Orion S. McCarthy.

**Visualization:** Orion S. McCarthy.

**Writing – original draft:** Orion S. McCarthy.

**Writing – review & editing:** Orion S. McCarthy, Morgan Winston Pomeroy, Jennifer E. Smith.

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
