## [Decision Letter · Decision Letter 0]

8 Mar 2024

PONE-D-24-02459Corals that survive repeated thermal stress show signs of selection and acclimatizationPLOS ONE

Dear Dr. McCarthy, Thank you for submitting your work to PLoS ONE. Although I normally make an effort to read the article myself, especially those such as these that address a topic of critical importance, I have unfortunately been too swamped as of recent. Thankfully, it appears that both reviewers were of the same mind: that this work should be published in PLoS ONE pending some minor revisions, so we look forward to seeing the revised article in the coming weeks. 

We look forward to receiving your revised manuscript.

Kind regards,

Anderson B. Mayfield, Ph.D.

Academic Editor

PLOS ONE

“We would like to thank the divers who helped conduct the large-area imagery surveys, including Anela Akiona, Donna Brown, Samantha Clements, Emily Kelly, Susan Kram, Travis Matteson, Melissa Torres, and Darla White. Special thanks to Emily Kelly for establishing this large-area imagery timeseries back in 2014. Field work would not have been possible without the support of local partners in Maui, including Ultimate Whale Watch, Dive Maui, Maui Divers, Lee James, Peter and Toni Colombo, Craig and Amy Venema, Don McLeish, Will and Megan Dailer, and George and Donna Brown. We would also like to thank undergraduate research volunteers Solomon Chang, Andre Lai, Veronika Pearson, Varun Sachin Shirhatti, and Victoria Vasquez for their tireless work tracing coral patches. We also thank Tom Oliver, Courtney Couch, and Mary Donovan, who provided input on project conception and initial results, and Adi Khen for designing the coral graphics used in our figures. Funding was provided to OM by the National Science Foundation, and to JES by the Scripps Family Foundation, the Bohn Family, and other generous donors.”

“Funding was provided to OM by the National Science Foundation (Graduate Research Fellowship Program award; https://www.nsfgrfp.org/), and to JES by the Scripps Family Foundation, the Bohn Family, and other generous donors. Funding was generalized and was not awarded for this project specifically. As such, funders had no role in study design, data collection, analysis, manuscript preparation, or publication.”

4. We note that Figures 1 and 2 in your submission contain copyrighted images. All PLOS content is published under the Creative Commons Attribution License (CC BY 4.0), which means that the manuscript, images, and Supporting Information files will be freely available online, and any third party is permitted to access, download, copy, distribute, and use these materials in any way, even commercially, with proper attribution. For more information, see our copyright guidelines: http://journals.plos.org/plosone/s/licenses-and-copyright.

1. You may seek permission from the original copyright holder of Figures 1 and 2 to publish the content specifically under the CC BY 4.0 license.

5. We note that Figures 1, 6, and Fig S5 in your submission contain [map/satellite] images which may be copyrighted. All PLOS content is published under the Creative Commons Attribution License (CC BY 4.0), which means that the manuscript, images, and Supporting Information files will be freely available online, and any third party is permitted to access, download, copy, distribute, and use these materials in any way, even commercially, with proper attribution. For these reasons, we cannot publish previously copyrighted maps or satellite images created using proprietary data, such as Google software (Google Maps, Street View, and Earth). For more information, see our copyright guidelines: http://journals.plos.org/plosone/s/licenses-and-copyright.

1. You may seek permission from the original copyright holder of Figures 1, 6, and Fig S5 to publish the content specifically under the CC BY 4.0 license. 

6. We notice that your supplementary figures are uploaded with the file type 'Figure'. Please amend the file type to 'Supporting Information'. Please ensure that each Supporting Information file has a legend listed in the manuscript after the references list.

Reviewers' comments:

Reviewer's Responses to Questions

**Comments to the Author**

1. Is the manuscript technically sound, and do the data support the conclusions?

Reviewer #1: Yes

Reviewer #2: Yes

2. Has the statistical analysis been performed appropriately and rigorously? 

Reviewer #1: Yes

Reviewer #2: Yes

3. Have the authors made all data underlying the findings in their manuscript fully available?

Reviewer #1: Yes

Reviewer #2: No

4. Is the manuscript presented in an intelligible fashion and written in standard English?

Reviewer #1: Yes

Reviewer #2: Yes

5. Review Comments to the Author

Reviewer #1: This study focusses on bleaching responses of nearly 2000 colonies of four coral species over seven years, overlapping two mass bleaching events in 2015 and 2019 at Maui. By examining bleaching extent, growth and survivorship of each coral through large-area photogrammetry, the authors found limited support for acclimatisation of populations within and among reefs. Individual colonies may be more tolerant and have lower bleaching extents over time, but there was generally no relationship between bleaching and coral growth.

The approach to track individual colonies is appropriate for testing acclimatisation at various scales—from colony to reef—and the application of photogrammetry over large areas does generate a much larger amount of data relative to conventional reef surveys. Taxonomic resolution may be limited, but it is sufficient for tracking four easily recognisable taxa. The text is well written and figures generally clear and support the results well, despite the poor resolution, likely a limitation of the manuscript submission system. I have a few comments and minor suggestions that hopefully will help clarify and improve the manuscript.

The coral colonies are holobionts that comprise the host, Symbiodiniaceae endosymbionts and microbial communities. While the characterisation of these components is understandably beyond the scope of this study, their contributions to thermal tolerance and resilience are significant and should be discussed.

The distinction between the effects of coral acclimatisation or taxonomic turnover on thermal tolerance at the reef level is also alluded to at various parts of the manuscript, but this does not appear to be tested specifically. Could the abundance changes between surveys be modelled to test for temporal variations in relative abundances of taxa and the effects on subsequent bleaching?

The inability to distinguish the Pocillopora spp. is slightly challenging for the inferences associated with population-level changes, primarily because these are masked by the corals being lumped as one taxonomic group. This limitation should be explicitly factored into the uncertainties surrounding results on Pocillopora.

A couple of minor issues on nomenclature: it should be scleractinian (word not capitalised) and Scleractinia (formal name with word capitalised); and do not capitalise family names, e.g. Symbiodiniaceae.

Abstract

This is well-written, but I thought the implications for restoration are not represented here. For example, the point that targeting specific phenotypes of Montipora or Porites based on thermal resistance may not lead to better outcomes should be mentioned here. Overall, the implications are written slightly too generally to be informative.

Lines 84–90

Rather than listing mechanisms and areas of study, it would be helpful to describe a couple of studies and their results as examples for understanding acclimatisation in corals.

Lines 127–130

Again here, a description of results of these studies would be constructive. In particular, why ‘but see Fukunaga et al. 2022’?

Furthermore, earlier it has been asked if changes in thermal tolerance could be driven by coral acclimatisation or taxonomic turnover. Could this be addressed with the analysis here?

Line 148

It would be clearer to have a temperature graph for each of the 5 years of surveys, indicating the survey date—something like Figure S2 but for all survey years, to have an understanding of the severity of the heat stress. And this should be in the main text.

Line 191

It’s not clear the necessary to mention ‘Researchers from both Scripps Institution of Oceanography (SIO) and Arizona State University (ASU) used the orthoprojections to delineate patches of live coral tissue.’

Line 210

Is this use of ‘patches’ problematic for defining genets when you have partially dead corals with patchy live tissue left on the original colony? Would separate ‘patches’ from the same colony (previously) be considered different genets, and does this become a pseudoreplication problem?

Line 232

‘Focusing on this bleached tissue only, we then scored bleaching severity (from 0 to 3) based on the overall degree of paling observed (0 for < 5% of pigmentation lost; 1 for 5 – 50%, 2 for 50 – 95%, and 3 for > 95%; Rodriguez et al. 2021).’

Is this measure based on colouration? What colour parameters are being used? E.g. what does 10% or 50% of paling mean? The current description is rather vague.

Line 319

forin?

Reviewer #2: The study performed by McCarthy et al., in my opinion is very interesting and well structured, as well as original and innovative in terms of methodology. All the technical and experimental procedures seem to have been performed in a precise and reliable way and the results showed are considerable in term of both quantity and quality. The work proposes some interesting findings that surely merit publication, also considering the constant increase in the intensity and frequency of bleaching events worldwide and the future challenges that corals will face in the context of global climate change. In addition, in my opinion, the main strong point of the work is the long monitoring period and the high number of colonies analysed. The paper is also well written.

For all these reason, I suggest to the Editor to accept the manuscript following a minor revision.

However, the main problem with this article is its length. Authors should cut some text in all the manuscript paragraphs and make the manuscript easier and more fluid to read. Many information, analysis and results are included in the manuscript, and the authors should try to make them more easily digestible for the reader.

Here, some specific comments section by section.

Introduction:

1) Pag. 3, Lines 64-65. “Still, scleractinian….(Pandolfi et al 2011). I would delete this sentence

2) Pag. 3, Lines 65-68. “More recently….still a possibility”. I would suggest to move this sentence after the line 79 (Pag. 4).

3) Pag. 4, Line 87. I would add as mechanism for acclimatization to heat stress also the coral physiological and molecular features (see for example Louis et al., 2020 Molecular Ecology, Bellantuono et al., 2012 PlosONE, Traylor-Knowles et al., 2017 JEB….).

4) Pag. 4 , Line 91. I would suggest to add also Maldives in the list (see Seveso et al. 2018, Coral Reefs)

5) Pag. 4, Line 96. I would delete the ref. Winston et al in prep throughout all the manuscript

6) Pag. 6, Lines 132-134. Since they are questions, why is the question mark missing?

Methods:

1) I would insert as figure a map of the study area in which also the different analyzed sites are indicated.

2) Pag. 7, Lines 167-168. I would add as Reference also Montalbetti et al 2022, Coral Reefs

3) Pag. 7, Lines 170-189. This part is too long. Please, shorten it.

4) Pag. 8, Lines 191-196. Is this information really necessary? If yes, try to write it differently

5) Pag. 10, Lines 244-247. I would delete the part in brackets.

Results:

In general, try to shorten all the result paragraphs (especially those related to Growth and Survivorship)

1) I would move the Fig. S4 in the Figure list of the manuscript

2) Pag. 13, Line 325. “…although the small sample size.”

3) Pag. 13, Lines 326-327. “Pocillopora…..event in 2015”. I would move this sentence to the Legend of Figure 4

4) Pag. 16, Line 394. Delete this sentence

Discussion:

The discussion is very exhaustive and well written, but I suggest to try to shorten it a bit (especially the paragraph related to coral restoration)

1) Pag. 17, Lines 410-411: use the past form for verbs (reflected, contributed..)

2) Pag. 17, Lines 414-415. “We were…..entire populations”. Delete this sentence

6. PLOS authors have the option to publish the peer review history of their article (what does this mean?). If published, this will include your full peer review and any attached files.

Reviewer #1: No

Reviewer #2: No

---

## [Author Response · Author response to Decision Letter 0]

14 Apr 2024

See attached "response to reviewers" document

---

## [Decision Letter · Decision Letter 1]

1 May 2024

Corals that survive repeated thermal stress show signs of selection and acclimatization

PONE-D-24-02459R1

Dear Dr. McCarthy,

We’re pleased to inform you that your manuscript has been judged scientifically suitable for publication and will be formally accepted for publication once it meets all outstanding technical requirements.

Kind regards,

Anderson B. Mayfield, Ph.D.

Academic Editor

PLOS ONE

Additional Editor Comments (optional):

Hello,

I am pleased to say that both reviewers have now endorsed your article for publication in PLoS ONE. The journal will be in touch in the coming days with proofs and things of that nature. Well done!

Anderson

Reviewers' comments:

Reviewer's Responses to Questions

**Comments to the Author**

1. If the authors have adequately addressed your comments raised in a previous round of review and you feel that this manuscript is now acceptable for publication, you may indicate that here to bypass the “Comments to the Author” section, enter your conflict of interest statement in the “Confidential to Editor” section, and submit your "Accept" recommendation.

Reviewer #1: All comments have been addressed

Reviewer #2: All comments have been addressed

2. Is the manuscript technically sound, and do the data support the conclusions?

Reviewer #1: Yes

Reviewer #2: Yes

3. Has the statistical analysis been performed appropriately and rigorously? 

Reviewer #1: Yes

Reviewer #2: Yes

4. Have the authors made all data underlying the findings in their manuscript fully available?

Reviewer #1: Yes

Reviewer #2: Yes

5. Is the manuscript presented in an intelligible fashion and written in standard English?

Reviewer #1: Yes

Reviewer #2: Yes

6. Review Comments to the Author

Reviewer #1: (No Response)

Reviewer #2: I would like to thank the authors for responding and addressing all my comments. I have no further information or suggestions to request. I think the article is now ready to be published

7. PLOS authors have the option to publish the peer review history of their article (what does this mean?). If published, this will include your full peer review and any attached files.

Reviewer #1: No

Reviewer #2: No
